# Neural Network-Aided Milk Somatic Cell Count Increase Prediction

**DOI:** 10.3390/vetsci12050420

**Published:** 2025-04-29

**Authors:** Sára Ágnes Nagy, István Csabai, Tamás Varga, Bettina Póth-Szebenyi, György Gábor, Norbert Solymosi

**Affiliations:** 1Department of Physics of Complex Systems, Eötvös Loránd University, 1117 Budapest, Hungary; nasarag@gmail.com (S.Á.N.); csabai@elte.hu (I.C.); 2Centre for Bioinformatics, University of Veterinary Medicine, 1078 Budapest, Hungary; vargatam1@gmail.com (T.V.); h12617gab@helka.iif.hu (G.G.); 3Doctoral School of Animal Science, Hungarian University of Agriculture and Life Sciences, 7400 Kaposvár, Hungary; poth.szebenyi.bettina@gmail.com

**Keywords:** dairy cow, subclinical mastitis, somatic cell count, electrical conductivity, machine learning, artificial neural network

## Abstract

Mastitis cause the biggest economical loss in the dairy industry worldwide. Its subclinical form, in the absence of visible symptoms, is difficult to diagnose under farm conditions without laboratory laboratory assistance. It is therefore particularly important to develop methods to reduce the incidence of subclinical mastitis in the herd to lower levels. Somatic cell count is the most commonly used method for monitoring subclinical mastitis. In our work we used artificial neural network to predict increased somatic cell count from data recorded by automatic milking machines, thus helping to detect subclinical mastitis at herd level. For this purpose, we used milk yield and milk-related data that can be made available to the owner by milking machines used in an average Hungarian dairy farm. Our best ANN model has a sensitivity of 0.54 and a specificity of 0.77, which exceeds the performance of the currently widely used diagnostic method performed cow-side, California Mastitis Test. Combined with the latter, the positive predictive value can be increased by a further 50%. This method may be a time-saving and cost-effective way to diagnose subclinical mastitis.

## 1. Introduction

Mastitis, or intramammary infection (IMI), is the most common infectious disease in the dairy cattle sector [1,2]. Its economic damage in the European Union is estimated at EUR 3 billion [3]. The financial impact comes on the one hand from reduced milk production [1,2,4] which can be as high as 0.3–1.9 kg/day of milk per cow [4]. On the other hand, the additional costs of early culling, the loss of revenue due to reduced milk quality, increased veterinary and pharmaceutical costs also reduce profitability [3]. Further costs are increased when the subclinical mastitis (SM) becomes clinical [5], which also leads to higher expenses [6]. In addition to the financial implications, increased antibiotic use due to subclinical mastitis also raises the problem of the spread of antibiotic resistance. The 60–70% of antibiotics used on dairy farms are for treatment of mastitis [2]. This opens up the possibility that with more efficient SM detection, more efficient antibiotic use could be achieved and even the development of clinical mastitis could be prevented [6]. Given that mastitis treatment accounts for a significant part of the dairy sector’s antimicrobial consumption, a good SM diagnostic system could be a great support for farmers to be able to meet the EU’s 20% antibiotic reduction target by 2030 [7].

Considering the significant economic and animal welfare and one health issues caused by mastitis [1,8], early detection and treatment of the affected animals is of utmost importance. Despite the 15–40 times higher appearance rates, SM, unlike clinical mastitis, does not cause visible symptoms in the udder or milk, but a decrease in milk production and quality [2]. As no clinical sign exists, additional tests are required for SM diagnosis. Even though bacterial testing and PCR are currently the best methods to diagnose IMI, they are expensive and time-consuming [9].

As an increase in the number of cells in milk is an indicator of an inflammatory response, the most commonly used method to detect IMI and assess udder health is to determine the total number of somatic cells in milk [8,9,10]. Somatic cell count (SCC) <200,000 cells/mL is considered healthy in individual case [2]. However, Ruegg and Pentoja [1] suggest that milk loss is seen as early as SCC 100,000 cells/mL. At the same time an SCC of 400,000 cells/mL is clearly considered an IMI. The use of SCC is limited because not only the presence of mastitis but other factors affect its results (e.g., age, season, diurnal variation, number of quarters with IMI) [1]. Furthermore, the frequent use of SCC determination as a diagnostic method is hampered by the fact that the execution requires trained experts and relatively high expenditure rates [10]. The most common laboratory cell counting method is based on staining cells in milk with fluorescent dye and counting the fluorescent particles with machine [11]. There is also a portable on-farm machine which can determine SCC. It also uses a fluoro-optical technique for cell counting, but does not require professional laboratory equipment [11]. A measurement method for SCC integrated in a milking robot is also possible. In the system described by Lusis et al, the milking robot deduces the SCC from the change in viscosity due to the addition of a reagent to the milk and the somatic cell reaction [12]. Currently, only a few conventional milking systems can continuously estimate SCC [8]. In contrast, conventional milking systems that measure the electrical conductivity of composite milk samples are widespread [8]. Although the change in the electrical conductivity of mastitis milk is a long-established phenomenon [13,14], its use for IMI detection is less common [8,10]. Furthermore, readily available indirect tests include the California Mastitis Test (CMT) [15] that provides approximate information on the quantity of somatic cells in milk by the agglutination of immune cells’ DNA in milk [16,17]. While CMT has the advantage of being simple to perform on single cow, its herd-level use is impractical. Moreover, CMT’s sensitivity and specificity are poor [16], and its evaluation is subjective [10].

The above-mentioned facts outline the need for a reliable, automatable, rapid SM detection method to reduce antimicrobial use and improve animal welfare parameters, the chance of recovery from mastitis, and economic indicators [8,10]. Several studies suggest that the combination of different indirect detection methods could yield better results in this field [9,16].

Machine learning models for predicting udder health are not unprecedented [6,18]. One part of the reason for this is that they are particularly well suited to biological datasets that are too large and complex [19]. Among machine learning models, artificial neural networks (ANNs) are also the most widely used in agricultural research and are also the most successful [20] ones.

In our work, we investigated how the combination of automatically collected data related to lactation and milk characteristics, accompanied by CMT parameters, could improve the prediction of SCC increase. For that, ANN were trained to classify SCC increase using available, automatically recorded parameters. Subsequently, the resulting ANN-based test was combined with CMT for the better performance.

## 2. Materials and Methods

### 2.1. Data Collection

Milk production data were collected from a large-scale Holstein Friesian herd in Hungary. The average milk yield is about 35 L/cow in milk daily. The herd of 850 milking cows are milked 3 times a day on a rotary milking parlour with 50 stalls. The milking machines collect data about milk and milk yield on cow level (e.g., blood traces in milk, conductivity, kick-off and air entry into liners). This data is detected by the sensors in the milking parlour and automatically registered by the software running the parlour (DeLaval ALPRO™, Stockholm, Sweden). The following data were collected: AverageFlow (average milk yield during milking, measured in kg/min), AvgBloodLevel (average amount of blood in the milk measured in ppm during milking), AvgCondLevel (average electrical conductivity of the milk during milking in mS), CowNo (the identification number of the cow), Duration (the length of the milking), GroupNumber (the identification of the group where the cow is found on the day of milking), HerdNo (the identification number of the herd), KickOff (sudden stop of milk flow during milking), MilkDateTime (the precise date and time of the milking), MPC (the identification number of the milking stall where the animal was milked), PeakBloodLevel (maximum amount of blood in the milk measured in ppm during milking), PeakCondLevel (maximum electrical conductivity of the milk during milking in mS), PeakFlow (maximum milk yield during milking, measured in kg/min), RelativeCond (expressing the change in electrical conductivity of the milk), Session (definition of the milking schedule: morning, midday and evening shift), SessionDateTime (date and exact time of milking shift), Yield (amount of yielded milk in kg), YieldIsLow (binary expression of the decrease in production compared to individual production) and YieldPercentage (milk yield relative to the individual’s own production).

On the farm, SCC measurements are taken once a month during a morning milking on all of the lactating animals that are not on antibiotic treatment. The SCC data is managed by another farm management software (Riska Farm Management System, Systo Ltd., Páty, Hungary) which contains various data on the complete history of the animal with unique identifiers for each individual as well. Here you can find all the information about the animal throughout its life, from its pedigree, body-weight measurements, moving between groups, reproductive performance, medications, diagnostic test results, hoof trimmings, milk production data and information on its removal from the herd also. In our study from this database we only used the data related to the monthly SCC measurement: date of the measurement, the measured SCC, days in milk when the measurement was taken, number of actual lactation, the identification number of the cow.

### 2.2. Data Preprocessing

The two datasets were linked via the unique identifier of the cows, and we filtered the measured data for morning milkings up to 3 days before the collection of SCC data for each individual. Further on, SCC values were used to create a binomial dependent variable, with a value of 1 if SCC was above 200,000 cells/mL and 0 below 200,000 cells/mL. Using the functions of the package caret [21], we filtered out correlated explanatory variables in R-environment [22] and estimated the variable importance using a binomial generalized linear model. For neural network training, we kept explanatory variables with variable importance values greater than 3.

In practice, an increase in SCC above 200,000 cells/mL is generally evaluated as a sign of SM. Therefore, we used the SM prevalence values reported in the literature to estimate the pre-test probability of increased SCC. Prevalence data from publications published after 2000 were included in our analyses, with a case definition of subclinical mastitis based on a cut-off of 200,000 cells/mL. The prevalence values from different countries range widely (Australia: 28.9% [23]; Brazil: 45.4–49.6% [24]; Finland: 19.0–22.3% [25]; Indonesia: 68.2% [26]). Accordingly, predictive values were estimated for a pre-test probability range of 0.19–0.68.

The classification bias of CMT shows considerable variability in the literature. Sensitivity and specificity values were selected using data from publications published after 2000 that met the following criteria SE and SP values were reported for the whole udder and not only for quarters. The cut-off value of 200,000 cells/mL was used in the case definition of SM. In addition, SE and SP values were reported for all intramammary infections, not just for minor or major infections. The following value pairs were included in the study: SE: 0.69, SP: 0.72 [27]; SE: 0.70, SP: 0.48 [28]; SE: 0.95, SP: 0.78 [29]; SE: 0.71, SP: 0.57 [16]; SE: 0.95, SP: 0.81 [30].

### 2.3. Evaluation Metrics

In our study, as in many similar works, SCC above 200,000 cells/mL was considered IMI even in the absence of clinical symptoms [31]. The quality of detection was assessed by the predictive efficiency of CMT, ANN trained by us, or a combination of these. Two measures we used for the evaluation are the negative predictive value (NPV = (1 − P) × SE/(P × (1 − SE) + (1 − P) × SP)) and the positive predictive value (PPV = P × SE/(P × SE + (1 − P) × (1 − SP))) [32]. In the formulas, P, SE, and SP are the pre-test probability, sensitivity and specificity of the test used, respectively.

### 2.4. Model Training

Artificial neural networks (Figure 1) represent a distinct category of machine learning with the structural similarity to biological neural networks [33]. The first ANNs were developed and applied based on the information processing and transmission model of neurons [34]. The ANNs basic building elements are artificial neurons [34]. An artificial neuron is a mathematical function that maps (converts) inputs into outputs in a defined way [34]. Starting from this simple unit, neural networks consist of a large number of neurons (Figure 1) arranged in at least three, but rather several layers [35]. The outputs of the neurons in one layer become the inputs of the next layer [36]. As the stimulus travels through the biological neurons, the initial input moves from artificial neuron to neuron along the structure of the ANN [19,36]. Each connection between neurons represents a learnable weight [37]. They are modified by the model during training. The learning process is actually the optimization of weights for a given task [36]. After the process of training on numerous data, the model basic aim is that generate meaningful output value(s) from a previously unknown input set [19].

In our work, the process can be translated as using data on milk and the cow producing it as the input information. Using this, we trained the neural network (i.e., optimized its weights) to obtain a prediction of the udder health status of the cow in question, whether or not she is affected by subclinical mastitis.

The dataset was split into two parts with a 70/30% split. The smaller part served as a test set. The larger part was also split into a training and validation set in a 70/30% ratio. The number of layers varied from 1 to 4, and the number of neurons per layer varied by increments of 64 between 64 and 512. ANNs were created using all possible combinations. The hidden layer of the model used the ReLu, while the output layer used the sigmoid activation function, as our output was binomial (increased/not increased SCC).

To find the best model, these ANNs were trained, and their classification performance was evaluated. We maximized the sensitivity during the training (50 epochs) while reducing the loss obtained in the validation dataset, with a specificity of 0.9 in the callback. After each epoch, if the sensitivity exceeded the previous maximum, the weights associated with the network were saved. ANNs were trained using TensorFlow [38] on a Tesla V100 32GB GPU. Finally, using the saved weights, we performed the classification on an NVIDIA GeForce P8 2GB GPU using the test set and used the model with the best F1 score (2TP/(2TP+FP+FN)) in the subsequent analyses.

### 2.5. Combination of Tests

When combining ANN and CMT in the analysis of SCC increase, the combined sensitivity (${\mathrm{SE}}_{parallel}$, ${\mathrm{SE}}_{serial}$) and specificity (${\mathrm{SP}}_{parallel}$, ${\mathrm{SP}}_{serial}$) change. Estimation of classification bias in parallel testing is: ${\mathrm{SE}}_{parallel}$ = 1 − (1 − ${\mathrm{SE}}_{ANN}$) × (1 − ${\mathrm{SE}}_{CMT}$), ${\mathrm{SP}}_{parallel}$ = ${\mathrm{SP}}_{ANN}$ × ${\mathrm{SP}}_{CMT}$. For sequential testing, it is: ${\mathrm{SE}}_{serial}$ = ${\mathrm{SE}}_{ANN}$ × ${\mathrm{SE}}_{CMT}$, ${\mathrm{SP}}_{serial}$ = 1 − (1 − ${\mathrm{SP}}_{ANN}$) × (1 − ${\mathrm{SP}}_{CMT}$) [39]. Where ${\mathrm{SE}}_{ANN}$ and ${\mathrm{SP}}_{ANN}$ are estimated from predictions on the test set using our trained ANN, and ${\mathrm{SE}}_{CMT}$ and ${\mathrm{SP}}_{CMT}$ are sensitivity and specificity, respectively, gathered from the literature [16,27,28,29,30].

## 3. Results

After the preprocessing of databases, a dataset containing 7685 records resulted. By importance filtering the following variables were kept for the modeling: PeakCondLevel, AvgCondLevel, RelativeCond, Yield, YieldIsLow, number of actual lactation, days in milk, and PeakCondLevel gauged on the 1st, 2nd and 3rd day before the SCC measurement day.

During the analysis, a model with 384 neurons in one hidden layer gave the highest F1 score (0.42 with ${\mathrm{SE}}_{ANN}$ = 0.54, ${\mathrm{SP}}_{ANN}$ = 0.77) on the test set among the ANNs tested with different architectures.

The SE and SP estimates of the CMT from different literature and our best ANN and the NPV and PPV obtained using 19–68% prevalence values are presented in the top two panels of Figure 2.

The other two rows in the figure show the combined predictions of ANN and CMT. The second row of panels in the figure shows the NPV and PPV estimates of the parallel combination, and the third row shows the NPV and PPV estimates of the serial combination plotted against the prevalence. The median and interquartile range (IQR) of the same estimates are summarized in Table 1, which describes the expected overall performance of tests and test combinations over a given range of prevalence.

Ranking the medians of these estimates in descending order, ANN was ranked 5th in the NPV and 3rd in the PPV order. As shown in Table 1, the serial combination gives higher PPV values than the parallel one, the former approach is more useful from the SM detection point of view.

To illustrate the predictive change that the serial combination of ANN and CMT can bring compared to CMT, the difference and the ratio of the NPV and PPV values obtained in both ways are shown in Figure 3. The median and IQR of the difference and ratio curves are reported in Table 2.

## 4. Discussion

Due to the damage caused by SM, it is usually a central issue in dairy farm management. Since SM diagnosis is mainly possible by measuring SCC, regular monitoring of this value is key to both profitability and animal welfare. However direct SCC measurement is not a common practice. The primary objective of our work was to investigate whether ANN can be used to improve the diagnosis of SM based on data that are available by using typical conventional milking systems.

In our study, we kept in mind the practical usefulness of the model, namely that it could also be suitable for SM monitoring. In order to reduce the presence of subclinical mastitis at herd level, a test with a high positive predictive value should be used which can effectively detect affected animals even at lower prevalence [40]. The well-known phenomenon that prevalence affects PPV and NPV can be observed in each sub-figure of Figure 2. As prevalence increases, PPV also increases along with it while NPV decreases. On the contrary, the NPV shows an increase with declining prevalence.

Higher PPV can be achieved by increasing the specificity of the test [41]. Accordingly, the neural network weights were varied during training to increase sensitivity while keeping the specificity high.

Improving SM detection with ANN is not unprecedented in the literature [9,42,43]. Machine learning models, including ANNs, based on milking data (electrical conductivity, lactose, milk volume, etc.) have been used for SM detection in research. The predictive values of the used models were generally found to be superior to those of traditional statistical approaches. For this reason, in addition to analyzing ANN solely, we also combined ANN prediction with CMT, a well-known and widely used indirect SM diagnostic method in practice. This combined approach we present was not found in the literature.

The prediction curves plotted in the second and third rows of Figure 2 show that the parallel combination of ANN and CMT resulted in an improvement in NPV. In contrast, the serial combination resulted in an improvement in PPV. As one can see in these sub-figures, a higher PPV can be obtained by increasing the specificity, but it would be preferable to achieve this in such a way that the sensitivity is as high as possible. Accordingly, we varied the neural network weights during training to increase the sensitivity by keeping the specificity value high. In general, the classification efficiency of models is evaluated by the area under (AUC) the receiver operating characteristic curve (ROC) value [9]. However, maximizing the AUC parameter does not mean maximizing specificity.

Among other properties, Figure 3 shows that, especially for low prevalence, the ANN+CMT combination significantly increases the reliability of the positive prediction. Compared to individual CMT, the serial combination resulted in a 55% increase in PPV at a prevalence of 20% and 39% at a prevalence of 30%. According to these results, the prediction performance of ANN is comparable to CMT for both predictive values. While implementing CMT is a labor-intensive process, we can obtain SCC increase predictions for all individuals on a daily basis in milking parlors where milk properties or milk yield data similar to those used in our study can be measured automatically. This does not mean the neural network and the weights we have trained can be used directly on all farms. Nevertheless, we believe the pipeline presented here can be adapted quickly in farms with similar source data.

The combination of various tests is a solution that is often applied in epidemiology to improve the predictive value of diagnostic tests. When interpreting the results obtained from a combination of tests, it is essential to note that the elemental assumption is that the tests used must be independent of each other. The two tests used in our work detect an increase in SCC on a different basis. CMT is based on higher amounts of DNA deriving from higher numbers of cells in milk, while ANN uses independent features of the milk. Since the two tests are thus uncorrelated, the predictions from their combination can be considered reasonable. A parallel or serial combination of two tests is possible. In the former case, the sensitivity of the combined test will be higher than that of the individual tests used; in the latter case, the specificity will be higher. For SM, this has been pointed out by other authors as well [16]. They used combinations of different tests for the indirect detection of IMI were evaluated. According to their conclusion, the combination of SCC and CMT or milk electrical conductivity only resulted in modest improvements in diagnostics compared to the use of CMT or milk electrical conductivity alone [16]. However, no data were found on how the combination of ANN and CMT changes the predictive values. Figure 2 shows that the combination of ANN and CMT improves the predictive values compared to individual tests. If, as with model selection, our goal in combining tests is to increase PPV, which we can achieve by increasing specificity, we can accomplish this by using serial testing.

Since pre-test probability is one of the parameters used to estimate predictive values, NPV and PPV depend on their value. Therefore, we used different prevalence values found in the literature in the estimations to show the extent of change in the predictive value in practically feasible circumstances. For the same reason, we have collected data from the literature on the diagnostic reliability of CMT. Although we searched the literature for prevalence, sensitivity, and specificity data that utilized SCC cut-off values of 200,000 cells/mL, there is heterogeneity in the formulation, evaluation, and presentation of the included studies’ results. Including other automatically measurable parameters (e.g., feed intake, activity) could complete and improve the presented model. Further studies are needed to test our approach in practice and compare the predictions with gold-standard methods.

## 5. Conclusions

For subclinical mastitis, the prediction reliability of the trained artificial neural network algorithm has reached the levels of the widely used but labor-intensive California Mastitis Test’s. Moreover, it is shown that the serial combination of the ANN and the CMT can significantly increase the positive predictive value on SM. Since the ANN uses data that is automatically registered with each milking, ANN-based pre-testing can be done on the whole herd every day. Animals found to be SM affected by ANN can be further evaluated by CMT testing to clarify their mastitis status. Thanks to the serial test combination, the predictive value of the method is increased, and at the same time, the labour and cost requirements can be reduced, as only one pre-screened batch of cows needs to be tested. Further research is required in order to investigate its practical applicability and performance. However our results suggest that relying on automated measurements can cost-effectively improve the early diagnosis of subclinical mastitis.

## Figures and Tables

**Figure 1 vetsci-12-00420-f001:**
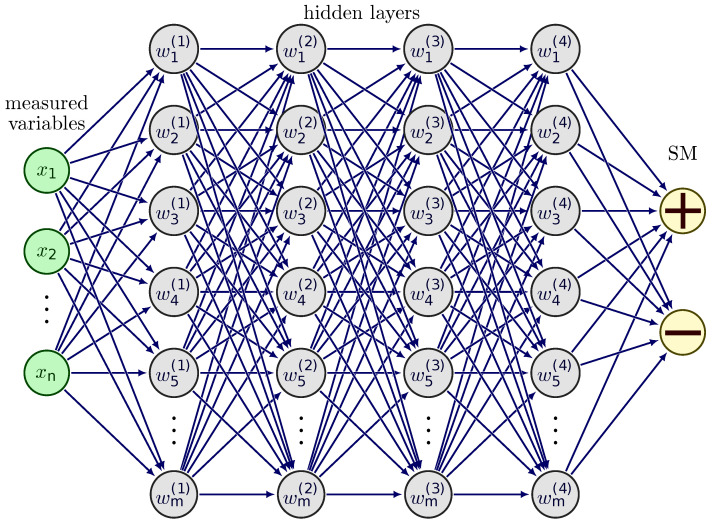
Fully connected artificial neural network scheme. The measured variables (*x*) stored in the databases used form the input layer, where the number (*n*) of nodes equals the number of the involved variables. The output layer (SM) contains nodes according to the target variable’s required class number. In our case, the subclinical mastitis state of a given animal can be positive or negative. Between the input and output layers, the nodes (*m*) of the hidden layers (upper index) store the parameters (weights: *w*) of the model. During the training, the weights are updated iteratively to reach the best classification of the output. The arrows represent the direction of information flow in training and prediction. From various architectures (with hidden layer and node numbers), based on the predictive performance, the best model is chosen by model selection.

**Figure 2 vetsci-12-00420-f002:**
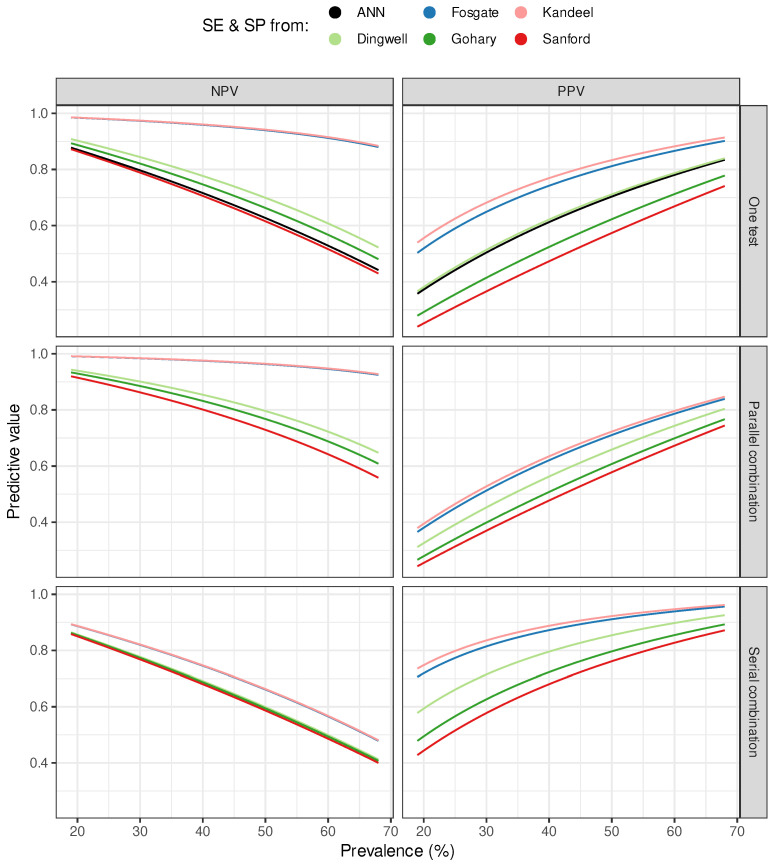
Predictive values. The black curve represents the ANN-based negative (NPV) and positive predictive values (PPV) as a function of subclinical mastitis prevalence. Based on sensitivity (SE) and specificity (SP) values of California Mastitis Test (CMT) published by Dingwell et al. [27], Fosgate et al. [29], Gohary et al. [16], Kandeel et al. [30], and Sanford et al. [28] estimated predictive values are shown by colored curves. In the first row of the figures (one test), the curves represent the predictive values using SE and SP of the individual tests (ANN or CMTs). The following two rows of figures show the predictive values obtained by combining ANN and CMT. The second row of figures shows their parallel combination, and the third row their serial combination.

**Figure 3 vetsci-12-00420-f003:**
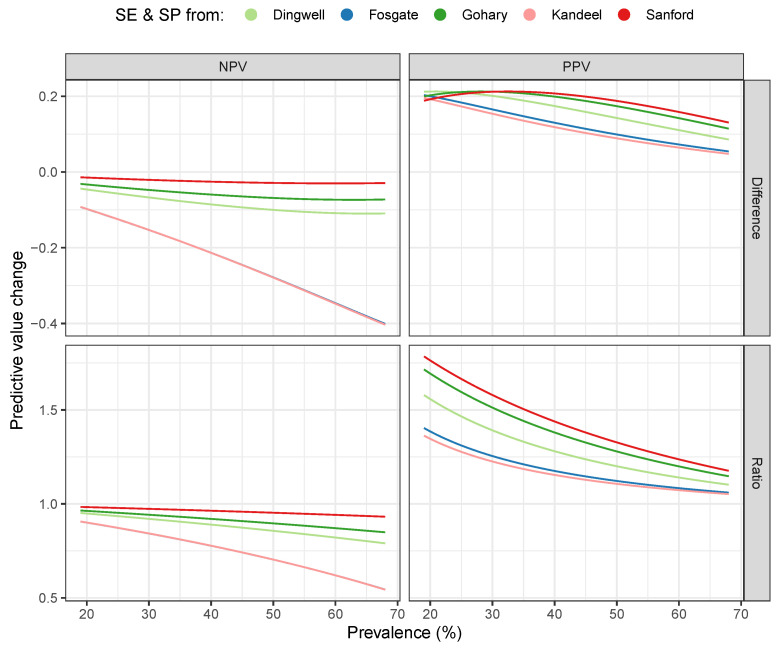
Prediction changes by serial combination. The curves show the predictive value changes of the ANN+CMT combination against CMT as a function of subclinical mastitis prevalence. The color of the curves indicates the literature source of the used CMT SE and SP values (Dingwell et al. [27], Fosgate et al. [29], Gohary et al. [16], Kandeel et al. [30], and Sanford et al. [28]). In the top row, the curves show the absolute difference between the ANN+CMT prediction values and the CMT prediction values, while in the bottom row, their ratios.

**Table 1 vetsci-12-00420-t001:** Descriptives of the negative and positive predictive values. For ANN, the NPV had a median 0.69 (IQR: 0.22), while PPV 0.65 (0.23). The predictive estimates based on the SE and SP values of CMT published by Dingwell et al. [27], Fosgate et al. [29], Gohary et al. [16], Kandeel et al. [30], and Sanford et al. [28] are in the One test columns. The Combination columns show the estimates of parallel and serial combinations of ANN and CMT testing.

Source	Negative Predictive Value			Positive Predictive Value		
	One Test	Combination		One Test	Combination	
		Parallel	Serial		Parallel	Serial
Dingwell	0.75 (0.19)	0.84 (0.14)	0.66 (0.22)	0.65 (0.23)	0.60 (0.24)	0.82 (0.15)
Fosgate	0.95 (0.05)	0.97 (0.03)	0.72 (0.20)	0.77 (0.18)	0.65 (0.23)	0.89 (0.10)
Gohary	0.72 (0.20)	0.81 (0.16)	0.66 (0.23)	0.56 (0.25)	0.54 (0.25)	0.75 (0.19)
Kandeel	0.95 (0.04)	0.97 (0.03)	0.72 (0.20)	0.79 (0.17)	0.67 (0.22)	0.90 (0.09)
Sanford	0.68 (0.22)	0.78 (0.17)	0.65 (0.23)	0.51 (0.25)	0.51 (0.25)	0.71 (0.21)

**Table 2 vetsci-12-00420-t002:** Descriptives of prediction changes by serial combination. The median and IQR of changes in predictive values between CMT and ANN+CMT serial combination estimates. The CMT test bias values were obtained from the paper of Dingwell et al. [27], Fosgate et al. [29], Gohary et al. [16], Kandeel et al. [30], and Sanford et al. [28].

Source	Difference		Ratio	
	NPV	PPV	NPV	PPV
Dingwell	−0.09 (0.04)	0.16 (0.07)	0.88 (0.08)	1.25 (0.21)
Fosgate	−0.24 (0.16)	0.12 (0.08)	0.75 (0.18)	1.15 (0.14)
Gohary	−0.06 (0.02)	0.19 (0.05)	0.91 (0.06)	1.34 (0.26)
Kandeel	−0.24 (0.16)	0.11 (0.07)	0.75 (0.18)	1.14 (0.13)
Sanford	−0.03 (0.01)	0.20 (0.04)	0.96 (0.03)	1.40 (0.29)

## Data Availability

The datasets used and analyzed during the current study are available from the corresponding author upon reasonable request.

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
