# Peer review of "Neural Network-Aided Milk Somatic Cell Count Increase Prediction"

_vetsci, 2025, doi:10.3390/vetsci12050420_

Round 1
Reviewer 1 Report
Comments and Suggestions for Authors
This paper has investigated whether the predictive value of SM diagnostics can be improved by training artificial neural networks (ANNs) on data generated using typical conventional milking systems. This work is meaningful for actual early detection and treatment of dairy cows. For improvement the paper, some issues should be addressed below.
- Please numbered the title and subtitle in this paper for better understand.
- Introduction, the background of the early detection and treatment of dairy cows should be addressed with more details, and the problem why need the ANN for this work, they all need to be addressed.
- The literatures such as the ANN, the methods for SCC and so on should be comprehensively addressed.
- Materials and Methods shoud add the basic methods for ANN and the models used in this paper.
- The experimetal schems fot the ANN could be added for better understand.
- Figure 1-2 are not clear, especial the curves addressed are not clear, so please improve the figures, and please numbered the figures inside.
- The results are not enough, only the figures 1-2 are simple, more results about the ANN performance and the results fore actual application for dairy cows should be added.
- Dicsuccsions should be added for discussing the results.
- The conclusions should added with the highlights and actual application extend in this paper.
- References should be updated with the newest, there are many old references such as 2013 2014 and so on.
Author Response
Comments: Please numbered the title and subtitle in this paper for better understand.
Response: Thank you for your suggestion, we have numbered the mentioned parts of the manuscript.
Comments: Introduction, the background of the early detection and treatment of dairy cows should be addressed with more details, and the problem why need the ANN for this work, they all need to be addressed.
Response: Thanks for your comment, we have rewritten and expanded our objective:
Mastitis, or intramammary infection (IMI), is the most common infectious disease in the dairy cattle sector [1,2]. Its economic damage in the European Union is estimated at EUR 3 billion [3]. The financial impact comes on the one hand from reduced milk production [1, 2, 4] which can be as high as 0.3-1.9 kg/day of milk per cow[ 4]. On the other hand, the additional costs of early culling, the loss of revenue due to reduced milk quality, increased veterinary and pharmaceutical costs also reduce profitability[ 3 ]. Further costs are increased when the subclinical mastitis (SM) becomes clinical [5], which also leads to higher expenses [6 ]. In addition to the financial implications, increased antibiotic use due to subclinical mastitis also raises the problem of the spread of antibiotic resistance. The 60-70% of antibiotics used on dairy farms are for treatment of mastitis [ 2]. This opens up the possibility that with more efficient SM detection, more efficient antibiotic use could be achieved and even the development of clinical mastitis could be prevented [6].Given that mastitis treatment accounts for a significant part of the dairy sector’s antimicrobial consumption, a good SM diagnostic system could be a great support for farmers to be able to meet the EU’s 20% antibiotic reduction target by 2030 [7].
Considering the significant economic and animal welfare issues caused [1 ,8], early mastitis detection and treatment is of utmost importance. Despite the 15-40 times higher appearance rates, SM, unlike clinical mastitis, does not cause visible symptoms in the udder or milk, but a decrease in milk production and quality [ 2]. As no clinical sign exists, additional tests are required for SM diagnosis. Even though bacterial testing and PCR are currently the best methods to diagnose IMI, they are expensive and time-consuming [9].
As an increase in the number of cells in milk is an indicator of an inflammatory response, the most commonly used method to detect IMI and assess udder health is to determine the total number of somatic cells in milk [ 8 –10 ]. Somatic cell count (SCC) <200,000 cells/mL is considered healthy in individual case [2 ]. However, Ruegg and Pentoja [ 1 ] suggest that milk loss is seen as early as SCC 100,000 cells/mL. At the same time an SCC of 400,000 cells/mL is clearly considered an IMI. The use of SCC is limited because not only the presence of mastitis but other factors affect its results (e.g. age, season, diurnal variation, number of quarters with IMI) [1 ]. Furthermore, the frequent use of SCC determination as a diagnostic method is hampered by the fact that the execution requires trained experts and relatively high expenditure rates [ 10]. The most common laboratory cell counting method is based on staining cells in milk with fluorescent dye and counting the fluorescent particles with machine [11]. There is also a portable on-farm machine which can determine SCC. It also uses a fluoro-optical technique for cell counting, but does not require professional laboratory equipment [11 ]. A measurement method for SCC integrated in a milking robot is also possible. In the system described by Lusis et al, the milking robot deduces the SCC from the change in viscosity due to the addition of a reagent to the milk and the somatic cell reaction [11].Currently, only a few conventional milking systems can continuously estimate SCC [ 8]. In contrast, conventional milking systems that measure the electrical conductivity of composite milk samples are widespread [8]. Although the change in the electrical conductivity of mastitis milk is a long-established phenomenon [13 , 14], its use for IMI detection is less common [8,10]. Furthermore, readily available indirect tests include the California Mastitis Test (CMT) [ 15 ] that provides approximate information on the quantity of somatic cells in milk by the agglutination of immune cells’ DNA in milk [ 16 ,17 ]. While CMT has the advantage of being simple to perform on single cow, its herd-level use is impractical. Moreover, CMT’s sensitivity and specificity are poor [16], and its evaluation is subjective [10].
The above-mentioned facts outline the need for a reliable, automatable, rapid SM detection method to reduce antimicrobial use and improve animal welfare parameters, the chance of recovery from mastitis, and economic indicators [8,10 ]. Several studies suggest that the combination of different indirect detection methods could yield better results in this field [9,16].
Machine learning models for predicting udder health are not unprecedented [ 6, 18 ]. One part of the reason for this is that they are particularly well suited to biological datasets that are too large and complex [19]. Among machine learning models, artificial neural networks (ANNs) are also the most widely used in agricultural research and are also the most successful [20] ones.
In our work, we investigated how the combination of automatically collected data related to lactation and milk characteristics, accompanied by CMT parameters, could improve the prediction of SCC increase. For that, ANN were trained to classify SCC increase using available, automatically recorded parameters. Subsequently, the resulting ANN-based test was combined with CMT for the better performance.
Comments: The literatures such as the ANN, the methods for SCC and so on should be comprehensively addressed.
Response: Thank you for the suggestion, we modified the text by this:
The most common laboratory cell counting method is based on staining cells in milk with fluorescent dye and counting the fluorescent particles with machine [10]. There is also a portable on-farm machine which can determine SCC. It also uses a fluoro-optical technique for cell counting, but does not require professional laboratory equipment [10 ]. A measurement method for SCC integrated in a milking robot is also possible. In the system described by Lusis et al, the milking robot deduces the SCC from the change in viscosity due to the addition of a reagent to the milk and the somatic cell reaction [11].
Comments: Materials and Methods shoud add the basic methods for ANN and the models used in this paper.
Response: Thank you for bringing this to our attention and we have expanded the methodology accordingly:
Artificial neural networks (Figure 1) represent a distinct category of machine learning with the structural similarity to biological neural networks [ 32]. The first ANNs were developed and applied based on the information processing and transmission model of neurons [33 ]. The ANNs basic building elements are artificial neurons [33 ]. An artificial neuron is a mathematical function that maps (converts) inputs into outputs in a defined way [ 33 ]. Starting from this simple unit, neural networks consist of a large number of neurons (Figure 1) arranged in at least three, but rather several layers [ 34]. The outputs of the neurons in one layer become the inputs of the next layer [ 35 ]. As the stimulus travels through the biological neurons, the initial input moves from artificial neuron to neuron along the structure of the ANN [ 18, 35 ]. Each connection between neurons represents a learnable weight [ 36 ]. They are modified by the model during training. The learning process is actually the optimization of weights for a given task [35]. After the process of training on numerous data, the model basic aim is that generate meaningful output value(s) from a previously unknown input set [18]. In our work, the process can be translated as using data on milk and the cow producing it as the input information. Using this, we trained the neural network (i.e., optimized its weights) to obtain a prediction of the udder health status of the cow in question, whether or not she is affected by subclinical mastitis.
Comments: The experimetal schems fot the ANN could be added for better understand.
Response: Thank you for the recommendation, we inserted a graph (Figure 1.) for the better understanding.
Comments: Figure 1-2 are not clear, especial the curves addressed are not clear, so please improve the figures, and please numbered the figures inside
Response: Thank you for the comment, we improved the figure explanation with compeletely rewritten caption. We haven't numbered inside the figures, because the panels have clear column and rownames.
Comments: The results are not enough, only the figures 1-2 are simple, more results about the ANN performance and the results fore actual application for dairy cows should be added.
Response: Thank you for your feedback. On the basis of that, we have added two new tables to the results to illustrate the significance of our findings. As we have only these results, we can put only those into the manuscript. But if the Referee could give us more detailed requirements to be presented we would be happy to provide those too.
Comments: Dicsuccsions should be added for discussing the results.
Response: Thanks for the observation, we modified the text accordingly. We separated the Results section from Discussion and also expanded them.
Comments: The conclusions should added with the highlights and actual application extend in this paper.
Response: Thank you for the comment. We added new aspects to the Conclusion section.
Comments: References should be updated with the newest, there are many old references such as 2013 2014 and so on.
Response: Thank you for your suggestion. We have updated the references with the latest possible citations related to the topic.
Reviewer 2 Report
Comments and Suggestions for Authors
Nicely writen and interesting field of research. I put some comments into the pdf.
I would suggest to better explain (in M&M) how ANNs are working in generell. Than it would be easier to unterstand for the readers.

Author Response
Comments: Nicely writen and interesting field of research. I put some comments into the pdf.
Response: Thank you for your comments. We have corrected the manuscript on the basis of those.
Comments: I would suggest to better explain (in M&M) how ANNs are working in generell. Than it would be easier to unterstand for the readers.
Response: Thank you for the suggestion. We extended the manuscript with the required ANN M&M description.
Reviewer 3 Report
Comments and Suggestions for Authors
The subject of SM is extremly important for dairy production world wide, so the subject of the manuscript is very actual.
Word "gain" in the title is not typically used in connection with SCC, mainy we are using "increase"
Abstract is clear, and gives all needed insides from the study. Just reviewer wont agree with statement from line 12, that "CMT is a labor-intensive process" - simple test, which even primarly trained personel can run in the barn. and Authors later in Introduction part mentioning it as "simply to perform" (line41).
Introduction part beggining looks like missing initial words (line 16). Line 18 - Authors can elaborate the antibiotics thread in the context of the EU target to reduce antibiotic consumption by 20% by 2030.
line 30 healthy for individual cow - please remember about EU reg. 853/2004 and bulk milk.
line 33 more details needed
Introduction needs some work effeorts of the Authors.
M&M please give more details about herd& animals used in the experiment.
results &Discussion
1st paragraph can be removed as it is a repetition from previous contents.
Fig. 1 &2 names by the color dots?
this part is barely related to literature references, and mainly concentrates on values - please think how to explain more for practitioners from dairy sector. Put more emphasis on conclusions.
please check all abbreviations used in the manuscript - if they were explained or make a list of abbreviations.
Manuscript could be improved to increse its value.
Author Response
Comments: Word "gain" in the title is not typically used in connection with SCC, mainy we are using "increase"
Response: Thank you for pointing out this, we have edited. And we have also replaced the gain words in the text in relation to SCC.
Comments: Abstract is clear, and gives all needed insides from the study. Just reviewer wont agree with statement from line 12, that "CMT is a labor-intensive process" - simple test, which even primarly trained personel can run in the barn. and Authors later in Introduction part mentioning it as "simply to perform" (line41).
Response: Thank you for highlighting the contradiction. In the original manuscript, the quote on line 12 refers to the authors' perception that it is labor-intensive to perform CMT at the herd level. We agree with the Reviewer that, of course, performing CMT at cow level cannot be considered labour-intensive. We have corrected the manuscript to avoid contradictions.
Comments: Introduction part beggining looks like missing initial words (line 16).
Response: Thank you for the observation. We corrected it.
Comments: Line 18 - Authors can elaborate the antibiotics thread in the context of the EU target to reduce antibiotic consumption by 20% by 2030.
Response: Thank you for your observation. This point of view has also been addressed in the process of improving the manuscript:
Given that mastitis treatment accounts for a significant part of the dairy sector’s antimicrobial consumption, a good SM diagnostic system could be a great support for farmers to be able to meet the EU’s 20% antibiotic reduction target by 2030 [7].
Comments: line 30 healthy for individual cow - please remember about EU reg. 853/2004 and bulk milk.
Response: Thank you for highlighting the unfortunate word choice. The authors aimed to prevent clinical mastitis by treating subclinical mastitis. But we completely agree with the reviewer that the wording was not correctly constructed in this regard, so we have removed it.
Comments: line 33 more details needed
Response: Thanks for your suggestion. We added more details in the aspect of SCC limitations:
The use of SCC is limited because not only the presence of mastitis but other factors affect its results (e.g. age, season, diurnal variation, number of quarters with IMI) [1 ].
Comments: Introduction needs some work effeorts of the Authors.
Response: Thanks for your comment, we have rewritten and expanded our objective: Please find the relevant part in the answer to question 2 of Reviewer 1.
Comments: M&M please give more details about herd& animals used in the experiment.
Response: Thanks for the suggestion. We extended that part with more information.
Comments: results &Discussion
1st paragraph can be removed as it is a repetition from previous contents.
Response: The Results and Discussion is splitted, and fundamentaly changed.
Comments: Fig. 1 &2 names by the color dots?
Response: Thanks for the highlight. The caption of the figures were rewritten with more extensive explanation.
Comments: this part is barely related to literature references, and mainly concentrates on values - please think how to explain more for practitioners from dairy sector. Put more emphasis on conclusions.
Response: Thanks for your opinion. The conclusion is extended by practical message.
Comments: please check all abbreviations used in the manuscript - if they were explained or make a list of abbreviations.
Response: Thank you for your suggestion, we have rechecked the abbreviations out of the abstract and explained the meaning at the first appearance of each.
Round 2
Reviewer 1 Report
Comments and Suggestions for Authors
Paper was revised well.